# Tuning Microelectrodes' Impedance to Improve Fast Ripples Recording

Hajar Mousavi [1,†], Gautier Dauly [2,†], Gabriel Dieuset [2], Amira El Merhie [1,3], Esma Ismailova [1,‡], Fabrice Wendling [2,‡] and Mariam Al Harrach [2,*,‡]

1   Bioelectronics Department, Ecoles des Mines de Saint Etienne, CMP-EMSE, MOC, 13541 Gardanne, France; seyedeh.mousavi@emse.fr (H.M.); amira.elmerhie@outlook.com (A.E.M.); ismailova@emse.fr (E.I.)
2   INSERM, LTSI-U1099, University of Rennes, 35000 Rennes, France; fabrice.wendling@univ-rennes1.fr (F.W.)
3   Laboratoire Matière et Systèmes Complexes, Université Paris Cité, CNRS UMR 7057, 10 Rue Alice Domon et Léonie Duquet, 75013 Paris, France
*   Correspondence: mariam.al-harrach@inserm.fr
†   These authors contributed equally to this work.
‡   These authors contributed equally to this work.

**Abstract:** Epilepsy is a chronic neurological disorder characterized by recurrent seizures resulting from abnormal neuronal hyperexcitability. In the case of pharmacoresistant epilepsy requiring resection surgery, the identification of the Epileptogenic Zone (EZ) is critical. Fast Ripples (FRs; 200–600 Hz) are one of the promising biomarkers that can aid in EZ delineation. However, recording FRs requires physically small electrodes. These microelectrodes suffer from high impedance, which significantly impacts FRs' observability and detection. In this study, we investigated the potential of a conductive polymer coating to enhance FR observability. We employed biophysical modeling to compare two types of microelectrodes: Gold (Au) and Au coated with the conductive polymer poly(3,4-ethylenedioxythiophene)-poly(styrene sulfonate) (Au/PEDOT:PSS). These electrodes were then implanted into the CA1 hippocampal neural network of epileptic mice to record FRs during epileptogenesis. The results showed that the polymer-coated electrodes had a two-order lower impedance as well as a higher transfer function amplitude and cut-off frequency. Consequently, FRs recorded with the PEDOT:PSS-coated microelectrode yielded significantly higher signal energy compared to the uncoated one. The PEDOT:PSS coating improved the observability of the recorded FRs and thus their detection. This work paves the way for the development of signal-specific microelectrode designs that allow for better targeting of pathological biomarkers.

**Keywords:** epilepsy; fast ripples; conducting polymers; microelectrodes; electrode–tissue interface

## 1. Introduction

Epilepsy is one of the most common neurological diseases that affects around 1% of the world's population [1]. In 30% of epilepsy cases, patients do not respond to available antiepileptic drugs [2]. For some of these patients, resective surgery can be considered as a potential treatment option [2,3]. However, a positive outcome of resection surgery highly depends on several factors, including accurately identifying the epileptogenic zone (EZ) during presurgical evaluation [4]. In that regard, the capacity to rely on objective biomarkers is fundamental to define the optimal surgical approach for each patient. Depth-EEG recordings performed with intracerebral electrodes are capable of recording local field potentials (LFPs) with a sub-millisecond temporal resolution. Fast ripples (FRs) are pathological high-frequency oscillations (200–600 Hz) observed in LFPs [5,6]. In the last two decades, they gained a vast interest in clinical applications as possible biomarkers of epileptic regions due to their high specificity [7–9].

FRs are mainly generated in primary sensory areas and hippocampal–entorhinal circuits in both humans and rodents [4,10,11]. They have been discovered to mirror the

underlying network changes during epileptogenesis, as they can be detected weeks before the onset of the first spontaneous seizure [2]. Many studies investigated the pathophysiological mechanisms of FRs' generation over the years. Recently, Al Harrach et al. [12] analyzed the evolution of FR during epileptogenesis using computational modeling and in vivo recordings from the CA1 subfield of the hippocampus of epileptic mice. They found that the generation of FRs is linked to two mechanisms: (a) the asynchronous firing of small clusters of pyramidal cells wherein each neuron fires at a frequency smaller than that of the recorded FRs; (b) hyper-excitability in the seizure onset zone, which can induce distant pathological plasticity in connected remote networks [12]. This statement is consistent with another study about the distinct hyperexcitable mechanisms underlying FRs' generation [13]

Clinical studies have not only highlighted the importance of FRs in the identification of the EZ, but more recently have shown a positive correlation between removing regions generating FRs and postoperative outcome [14–16]. However, the accurate detection of FRs remains a challenge due to their short duration, non-stationary behavior, and low amplitude that are often mixed with background activity and noise [17–19]. Microelectrodes are extensively used to record pathological fast ripples (FRs) due to their ability to provide high spatial resolution and selectivity, allowing for the recording of local field potentials (LFPs) from a smaller population of neurons [17,20]. However, they suffer from high impedance caused by their small size, particularly in frequencies below 1 kHz [21]. Microelectrode's impedimetric profile affects the recording of signals. First, the signal-to-noise ratio (SNR) level decreases due to the higher contribution of thermal noise [22,23]. Second, microelectrodes will have a high cut-off frequency of around 10 kHz. Beyond this threshold, the impedance increases by several folds due to the gradual transformation of the electrode impedance from a resistive to a capacitive regime [21]. This results in phase-shift and the non-linear distortion of signals, which can significantly affect the quality of the recorded FRs [17,24].

Over the past few years, several high-performance microelectrode designs have been proposed with the aim of optimizing the neural recording. These designs entail a tradeoff between achieving high electrical performance and maintaining adequate spatial resolution [17,25,26]. In particular, the introduction of coating microelectrodes with conductive polymers (CP) has proven to be a successful strategy for reducing the impedance and cut-off frequency of microelectrodes [27,28]. CPs are mixed ion/electronic conductors that enhance communication between ionically conducting living tissue and electronically conductive electrodes. The permeability of CPs to ions is crucial for reducing impedance and the cut-off frequency, as it enables electrochemical interaction throughout the entire bulk material [29,30].

Among conductive polymers, poly(3,4-ethylenedioxythiophene)-poly(styrene sulfonate) (PEDOT:PSS) is commonly used because of its thermal, electrochemical, and moisture stability, optical transparency, low oxidation potential, and commercial availability [27,28]. It promises to improve signal transfer while maintaining the geometrical size unchanged [17]. PEDOT:PSS is synthesized via the oxidation of EDOT (3,4-ethylenedioxythiophene), in the presence of PSS counter ions [27]. Its deposition on microelectrodes reduces the impedance by two orders of magnitude and pushes the cut-off frequency to a few tens of hertz thanks to its volumetric capacitive behavior [28,31].

Several studies have analyzed the impact of impedance on recorded signal quality focusing on LFPs and individual neuron activities. However, aside from our recent study about the model-guided design of microelectrodes to improve FRs' recording [17], no other studies have investigated the impact of microelectrode impedance on the transfer function of the recording system and its direct impact on FRs' observability. In our previous study, we used in silico modeling to test FR observability with different types of microelectrodes made with different materials and combined with different coatings [17]. We concluded that using PEDOT:PSS-coated microelectrodes can improve FRs' observability [17]. There was concern, however, that the results were only based on one day of in vivo recordings. In this

study, we investigate the effect of the tuning of a Gold (Au) microelectrode impedance via the coating with PEDOT:PSS (Au/PEDOT:PSS) and its impact on FRs' observability during epileptogenesis using a Kainate mouse model of temporal lobe epilepsy. To quantify the impact of the PEDOT:PSS coating, we developed equivalent circuit models and compared the observed FRs' energy feature contents for Au and Au/PEDOT:PSS microelectrodes.

## 2. Materials and Methods

### 2.1. Electrode Preparation

Gold wires with a diameter of 0.125 mm, insulated by polyester, were procured from the Goodfellow company. A pair of wires was joined together using super glue and insulated with a 2 μm layer of Parylene-C (applied using the PDS 2010 CSC deposition system). The electrode tips were delicately wet-polished on silicon 2000 paper at 50 rpm using the Structure Labopol-5 to expose the recording sides and subsequently soldered to the connector (obtained from Digikey, part number 850-10-050-10-001000). PEDOT:PSS was electropolymerized on the electrodes using an aqueous solution containing 0.01 M 3,4-ethylenedioxythiophene (EDOT) and 0.1 M Sodium Polystyrene Sulfonate (NaPSS) under potentiostatic conditions at 1.1 V for a duration of 50 s (Metrohm Autolab B.V.) in a three-electrode set up with an Ag/AgCl electrode as the reference and platinum wire as the counter electrode [32]. An Example of the Au and PEDOT:PSS coated Au electrode is depicted in Figure 1.

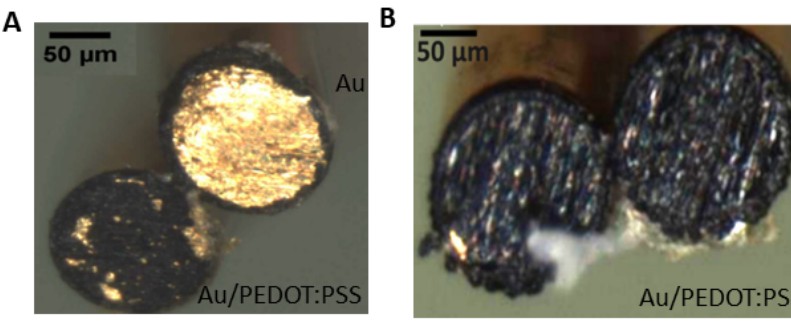

**Figure 1.** Microelectrodes' preparation and coating. Scanning electron microscope (SEM) images of the cross section of gold (Au) and Au coated with PEDOT:PSS (Au:PEDOT/PSS) wire microelectrodes implanted into the right (**A**) and left hippocampus (**B**) of the mouse. All electrodes have the same diameter of 125 μm.

### 2.2. Electrodes Modeling

Impedance measurements were conducted potentiostatically at a voltage of 0.01 V in a three-electrode set up (the Ag/AgCl electrode as the reference and platinum wire as a counter electrode), spanning the frequency range from 1 Hz to 10 kHz. The Bode plot of the Au and Au/PEDOT:PSS microelectrodes are depicted in Figure 2A. The mean values averaged for the five electrodes of each type (coated and uncoated) are given in Table 1. These measurements were modeled using equivalent circuits in Nova 1.2 software (Metrohm Autolab B.V.) to assess the electrochemical characteristics of the interface. The chosen circuits were adapted from the Randles circuit and modified based on the surface coating. Specifically, we introduced a model for the resistance to current flow induced by ion migration, represented by the spreading resistivity ($R_S$). This component describes the resistance between the working and the counter electrode and varies based on the size of the electrode and the resistivity of the interfacing solution. The non-faradic portion of the current in impedance measurements was quantified by the double-layer capacitance ($C_{dl}$). For faradic current modeling, we incorporated a charge transfer resistivity ($R_{ct}$), a Warburg element ($W$), and a constant phase element ($Q$). $W$ reflects the impedance dependence on frequency, showing the diffusion of solution species towards and away from the electrode–electrolyte interface due to a gradient in ionic concentration. $Q$ models

the surface imperfections and the porosity of the interface, gauging the proximity of the interface to an ideal capacitor [32,33]. The surface morphology of the electrode was imaged before and after the PEDOT:PSS coating using scanning electron microscopy (SEM) (Carl Zeiss Ultra55). The images were captured with the secondary electron detector (SE) at 5 kV (Figure 1A,B).

### 2.3. Cyclic Voltammetry Measurements

The volumetric capacitance of the electrodes before and after coating with PEDOT:PSS was measured using the cyclic voltammetry (CV) technique in a three-electrode set up (The Ag/AgCl electrode as the reference and platinum wire as a counter electrode). The CV scans are applied with scan rates of 100 mV/s in a voltage range of −0.9 to 0.6 V in an electrolyte solution. The aqueous solution used consisted of a Phosphate-Buffered Saline (PBS) of 0.01 M phosphate buffer, 0.0027 M potassium chloride, and 0.137 M sodium chloride (pH 7.4, at 25 °C).

**Table 1.** The mean values of the equivalent circuit elements used in the ETI model for the Gold (Au) and PEDOT/PSS-coated Gold (Au/PEDOT:PSS) electrodes of 125 μm diameter.

| Circuit Elements | Au | Au/PEDOT:PSS |
|:---:|:---:|:---:|
| $R_S$ (Ω) | $3 \times 10^3$ | $3 \times 10^3$ |
| $R_{CT}$ (Ω) | $12.24 \times 10^3$ | — |
| $C_{dl}$ (F) | $1.036 \times 10^{-9}$ | $1.73 \times 10^{-6}$ |
| $Q$ (F) | $10.97 \times 10^{-9}$ | — |
| $W(\Omega.s^{-0.5})$ | — | $10.24 \times 10^3$ |

Au: gold electrode, Au/PEDOT:PSS: PEDOT:PSS-coated gold electrode

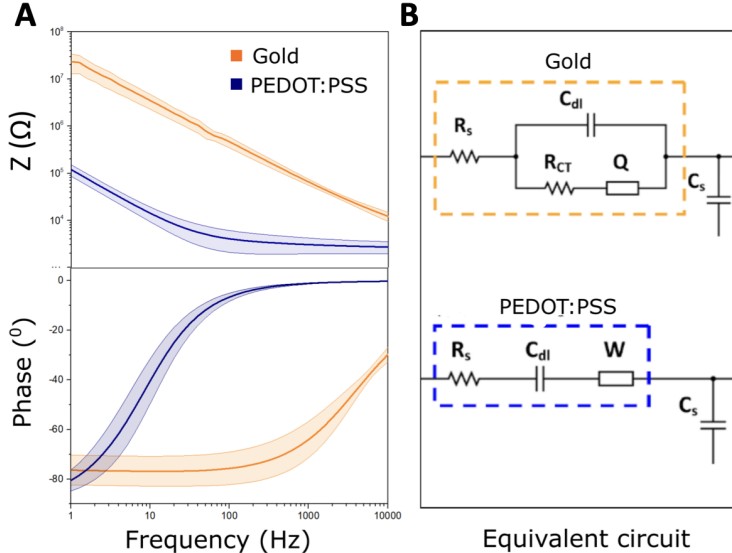

**Figure 2.** Electrode- Tissue Interface (ETI) model. (**A**) Bode plot representation of experimentally measured impedance using electrochemical impedance spectroscopy for gold (Au) and PEDOT:PSS-coated Au (Au/PEDOT:PSS) microelectrodes. (**B**) Equivalent circuits for the Au (yellow) and Au/PEDOT:PSS (blue) microelectrodes. For the Au electrode, the equivalent circuit consisted of the spreading resistance ($R_s$), the charge transfer resistance ($R_{CT}$), a constant phase element ($Q$), and double-layer capacitance ($C_{dl}$). For the Au/PEDOT:PSS, the equivalent circuit comprised a spreading resistance ($R_s$), double-layer capacitance ($C_{dl}$), and a Warburg element $W$. For both electrodes, the ETI included a shunt capacitance ($C_s$) that followed in series with the equivalent circuit of the electrode.

*2.4. Transfer Function Definition*

Based on the equivalent circuits of the Electrode–Tissue Interface (ETI) presented in Figure 2B, the transfer function of the system $H(j\omega)$ defined by the ratio between the input and output voltage ($H(j\omega) = \frac{V_{out}}{V_{in}}$) is given by Equation (1). The resistivity of the cables and connectors was assumed negligible. $H(j\omega)$ has the form of a voltage divider before the amplifier [21,24].

$$H(j\omega) = \frac{1}{j\omega C_s Z_{electrode} + 1} \tag{1}$$

Depending on the coating, the impedance of the electrode $Z_{electrode}$ can either follow Equation (2) for Au or Equation (3) for Au/PEDOT:PSS electrodes.

$$Z_{electrode}(j\omega) = R_s + \frac{1}{j\omega C_{dl} + \dfrac{1}{R_{CT} + Q}} \tag{2}$$

$$Z_{electrode}(j\omega) = R_s + \frac{1}{j\omega C_{dl}} + W \tag{3}$$

Additionally, the system is composed of connectors and wires to the amplifier. It is shown by a shunt Capacitance $C_s$, depicted on Equation (4). It is composed of a parasitic capacitive effect from the microwires being emerged in the medium, usually a few pF [34]. The connectors, wires, and amplifier add another capacitance, which can range from 10 pF to 100 nF. In the presented ETI model, the $C_s$ was considered to be equal to 3 nF [24,35].

$$C_s = C_{wires,connector} + C_{amplifier} \tag{4}$$

*2.5. Experimental Recordings*

An Intracerebral ElectroEncephalography (iEEG) recording of freely moving animals was performed on a set of five ($n = 5$) $C57BL/6JRj$ male mice of 8 to 12 weeks old in accordance with the Kainate mouse model of epilepsy [36]. This animal model was chosen for homology with human mesial temporal lobe epilepsy (MTLE) [37]. During the surgery, anesthetized and analgesized animals are positioned in a stereotaxic frame (Figure 3A). 50 nL of a 20 mM solution of kainic acid (KA; Sigma-Aldrich) in 0.9% NaCl is injected in the dentate gyrus (DG) of the right hippocampus (RH), at the coordinates AP = −2.0 mm, ML = −1.5 mm, and DV = −2.0 mm (all coordinates from bregma), via a cannula of 0.2 mm of diameter.

Two pairs of wire electrodes were implanted in the right (RH) and left (LH) hippocampus, above the DG region (Figure 3B). The first pair of electrodes consisted of one Au and one Au/PEDOT:PSS wire glued together and inserted in the RH (AP = −2.0 mm, ML = −1.5 mm, and DV = −2 mm). The second pair consisted of two Au/PEDOT:PSS microelectrodes and was implanted in the LH (AP = −2.0 mm, ML = +1.5 mm, and DV = −2 mm). The electrodes' placements were determined according to the atlas of the mouse brain ("Paxinos and Franklin's the Mouse Brain in Stereotaxic Coordinates" [38]). Finally, a stainless steel (SS) electrode was inserted in the skull above the cerebellum as a reference. The wires were all soldered to a connector fixed to the mouse's skull via dental acrylic cement. Recording sessions of iEEG were performed on days 2, 4, 7, 9, and 11 post-implantation (Figure 3C). During these sessions, the mice were placed in individual transparent cages inside a Faraday cage and connected to the recording system (Deltamed TM) for two hours at a sampling frequency of 2048 Hz. The initial hour is dedicated to habituation and is not taken into account during subsequent signal processing. This experimental procedure respected the European Union directive in use (Dir 2010/63/UE) and was approved by the ethics committee on animal experimentation of Rennes and received agreement from the French national legal entities (agreement N APAFIS # 7872-2017031711448150).

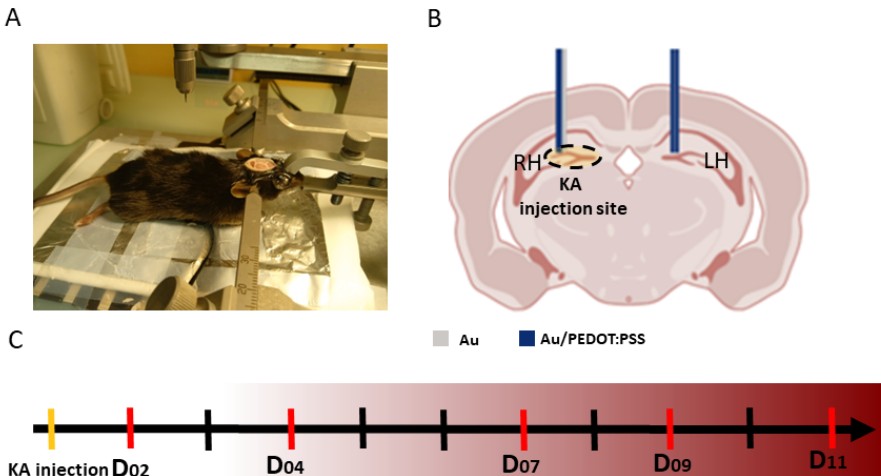

**Figure 3.** Experimental protocol setup. (**A**): Schematic diagram of the multisite intracortical electrode implantation positions. In the Right Hippocampus (RH), one Au and Au/PEDOT:PSS electrode each were inserted at AP = −2.0 mm, ML = −1.5 mm, and DV = −1.9 mm. In the Left Hippocampus (LH), two Au/PEDOT:PSS electrodes were inserted at AP = −2.0 mm, ML = +1.5 mm, and DV = −1.9 mm. The electrodes' placements were determined according to the atlas of the mouse brain [38]. (**B**): An image of the operating field during electrode implantation for one of the mice. The mouse is fixed in a stereotaxic frame. (**C**): Timeline of experimental design indicating recording days. Abbreviations: D: day post-injection, LH: left hippocampus, RH: right hippocampus, KA: kainate acid.

### 2.6. FRs' Identification and Analysis

The FRs underwent manual classification using time and frequency criteria. Events were designated as true FRs if they adhered to the following conditions: (i) including at least four clear oscillations in the FRs band (200–600 Hz); (ii) having, as the amplitude of an oscillation, at least twice the amplitude of the background; and (iii) evoking a well-defined spot on the spectrogram that is not a harmonic of lower frequency oscillations like Ripples (R; 120–200 Hz). These benchmarks are critical since FRs are easily mistaken for noise and artifacts due to the presence of many sharp events. In addition, the band-pass filtering of FRs is misleading since it can lead to misinterpretation of "false-Ripples" [39].

The manual classification process consisted of several steps that are portrayed in Figure 4A. First, a spectral decomposition was performed using a convolution between the signal and Gabor wavelets (Adapted from Ref. [40]). Eight Gabor functions are defined to decompose the signal on a filter bank defined by the following frequency bands: $\delta$ [0.5–3.5 Hz], $\theta$ [3.5–8 Hz], $\alpha$ (8–15 Hz), $\beta$ (15–30 Hz), $\gamma$ (30–80 Hz), High-$\gamma$ (80–120 Hz), R (120–200 Hz), and FR [200–600 Hz]. The background activity was defined as the signal recorded in the frequency band ranging from $\theta$ to R (3.5–200 Hz). Then, after visual pre-selection and classification, the time frames identified as true FR contained the high-frequency event itself, along with the activity before and after, as depicted in Figure 4B. Finally, the segmentation of the FRs to determine the time index of the onset and offset was carried out using the algorithm outlined in [12]. An example of the segmentation results are displayed in Figure 4C,D in the frequency and time domains, respectively. Figure 4C depicts the watershed segmentation algorithm result for an event of interest. It portrays the spectrogram of the signal in the FR band. The algorithm allows for the delineation of the FR onset and offset. This delineation is portrayed in Figure 4D in the time domain. The time stamps (onset and offset) of FRs were used to extract the specific energy features to compare between electrode types. This comparison was made by comparing the energy of the same recorded FR in the FR and Background bands, defined by Equations (5) and (6).

$$Energy_{FR} = \sum_{i=onset}^{offset} X_{FR}[i]^2 \tag{5}$$

$$Energy_{Background} = \sum_{i=onset}^{offset} X_{Background}[i]^2 \tag{6}$$

where $X_{FR}$ and $X_{Background}$ refer to the same signal in the FR and Background bands, respectively.

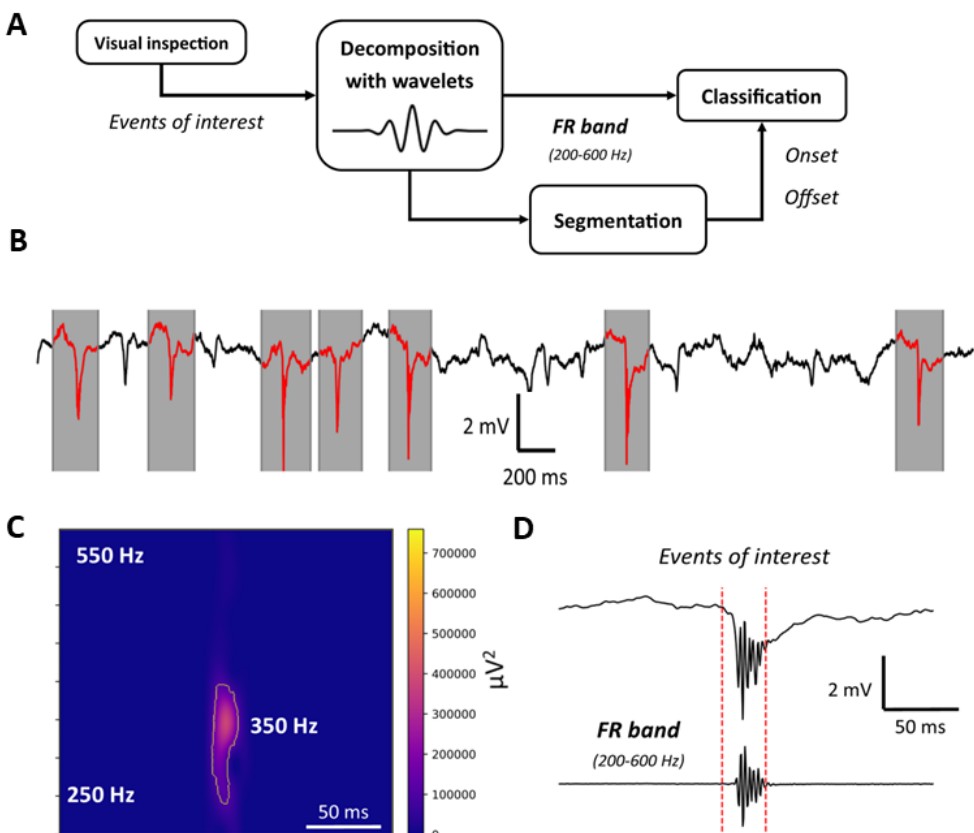

**Figure 4.** FRs' classification process. (**A**): Bloc diagram detailing the steps of the classification. (**B**): Visual inspection and delineation of the signal. FRs are highlighted in red. (**C**): Spectrogram of the event of interest segmented with the algorithm in [12]. (**D**): An example of FRs' segmentation in the time domain. The onset and offset of the FR are marked by red-dotted vertical lines for the unfiltered (top) and filtered (bottom) signal.

## 3. Results

In this study, we evaluated the impact of the microelectrode impedance on the quality of the recorded FRs. Accordingly, we coated standard Au wire electrodes with PEDOT:PSS to study the relation between impedance tuning and FR observability. The electrodes were prepared in pairs as portrayed in Figure 1. The SEM micrographs, presented in Figure 5A, show the morphology of the electrode's surface before and after the PEDOT:PSS coating. The resulting PEDOT:PSS roughness is due to the surface structure of the Au electrode and its initial roughness. Electrochemical impedance measurement (EIS) was employed to verify the performance of the fabricated electrodes. After coating the Au electrodes with PEDOT:PSS, as predicted, the impedance dropped by two orders of magnitude (Figure 2A). In particular, at 500 Hz, it decreased from 110 $\Omega$ ± 1.5 k$\Omega$ to 1.612 ± 0.2 k$\Omega$. This decrease is due to the PEDOT:PSS volumetric capacitive behavior [28,29]. To characterize the PE-DOT:PSS coating, we used the CV technique in the same electrochemical cell. Figure 5B shows the CV curve of the Au/PEDOT:PSS microelectrode obtained in a PBS solution. After 400 consecutive oxidation–reduction cycles at a scan rate of 100 mV/s [33], the PEDOT:PSS electrochemical properties remained unaltered. Lastly, we measured the impedance varia-

tion at 1 kHz over 4 weeks. The impedance magnitude plot is depicted in Figure 5C in kΩ. We observed a slight increase in the impedance over the first week, then it decreased to its original value in the following couple of weeks to settle at 2.25 KΩ throughout the last week of monitoring, indicating its stability (Figure 5C).

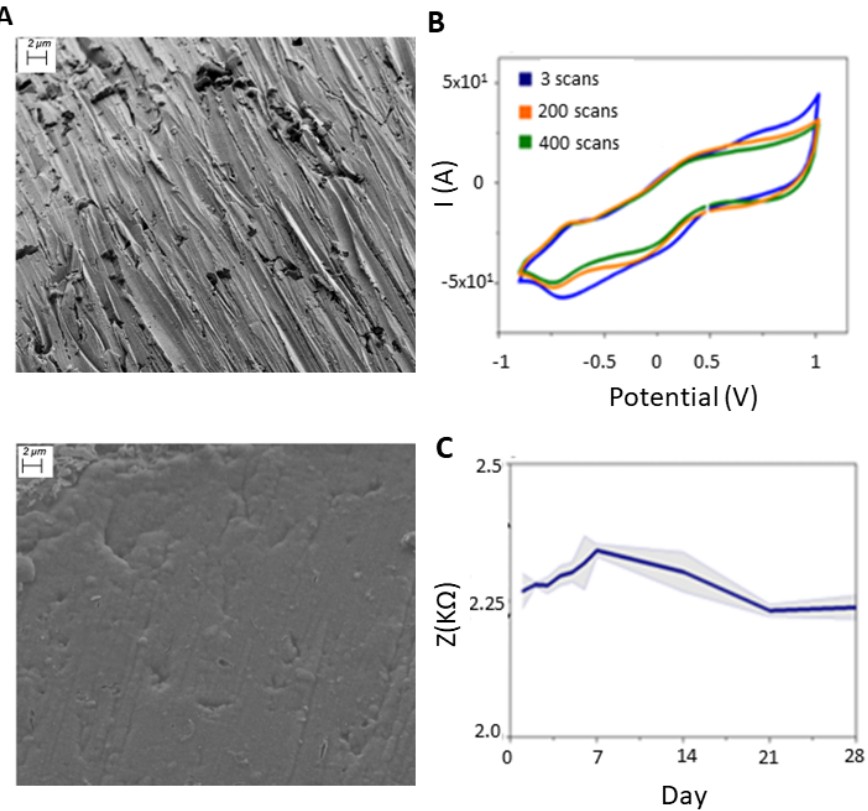

**Figure 5.** Coating characterization and electrochemical stability of PEDOT: PSS. (**A**): SEM images of the Au microelectrode surface before (up) and after (bottom) coating with PEDOT: PSS. The images were captured with the secondary electron detector (SE, Carl Zeiss Ultra55) at 5 kVA. (**B**): Cyclic voltammetry (CV) of Au/PEDOT: PSS microelectrode (100 mV/s, −0.9 to 1 V). (**C**): 1-KHz impedance of Au/PEDOT: PSS variation during 28 days.

To evaluate the change in the cut-off frequency after coating with PEDOT:PSS, we analyzed the equivalent circuit transfer function (H($jw$)) variation. Figure 6 depicts the transfer functions of the Au and Au/PEDOT:PSS microelectrodes in Bode plot form. As predicted, the transfer function of the system (Figure 6) predicts a low-pass filtering effect on the signal, where the shunt capacitance tunes the cut-off frequency. For the Au electrode (without coating), the transfer function does not present a typical profile (Figure 6). There is no predominant capacitive or resistive profile. From 1 Hz to 3 kHz, the slope of the attenuation is −1.1 dB/dec, starting at −3.12 dB. After this point, the slope becomes sharper. At 400 Hz, in the middle of the FR band, the gain of the Transfer function is around −12 dB for $C_s$ = 1 nF. In the case of coated electrodes, the cut-off frequency of the ETI seems to be at the end of the FR band (Figure 6). Based on the transfer function phase variation, in the FR and lower frequency bands, the phase of the signal should not be altered. The improvement in the filtering effect of the microelectrode after coating is directly related to the decrease in the impedance. The PEDOT:PSS coating shifted the capacitive behavior of the electrodes to a frequency as low as 63 ± 0.1 Hz (Figure 2B). This is reflected in the transfer function cut-off frequency variation due to the electrode's transition from resistive to capacitive behavior [21].

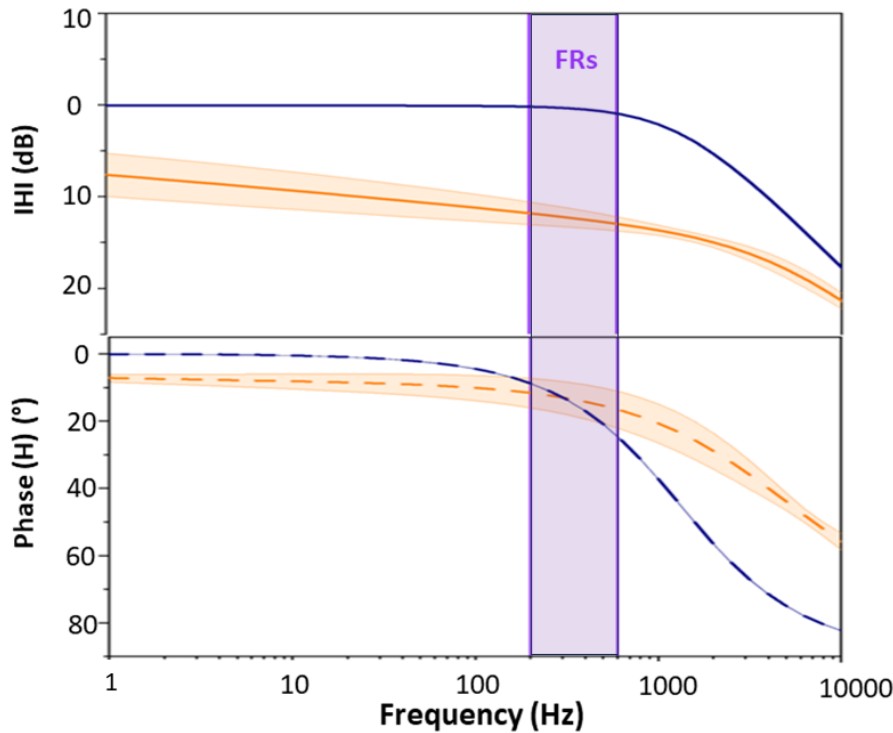

**Figure 6.** Bode plot representation of the transfer function (H) containing magnitude (logarithmic) and phase (linear) for Au and Au/PEDOT:PSS electrodes. The FR band is represented vertically (between 200 and 600 Hz).

From an electrochemical point of view, the spread resistivity ($R_s$) remained unchanged before and after the PEDOT:PSS coating at $2 \pm 0.1$ kΩ since it only depends on the electrode's geometrical size. $C_{dl}$ increased by three orders of magnitude from $6 \pm 0.5$ nF to $1.32 \pm 0.12$ µF as PEDOT:PSS is deposited. This is due to the increase in the effective electrochemical surface area of the microelectrode after coating. $R_{ct}$ rises from $1$ µΩ $\pm 0.01$ to several GΩ after PEDOT:PSS deposition, which signifies that the polymeric film acted as an ideal capacitor and hindered charge transfer at the electrode–electrolyte interface. So, $R_{ct}$ can be safely removed from the circuit for the Au/PEDOT:PSS electrode. Due to the rough surface of the Au wires before coating with PEDOT:PSS, we used the constant phase element which accounts for imperfections on the surface with the n value around 0.890, which is due to the surface roughness as shown in SEM image (Figure 5A). After coating, the n value of the constant phase element is almost 1, which reflects the capacitive behavior of PEDOT:PSS. Therefore, for PEDOT:PSS-coated electrodes, we replaced the Q with the Warburg element to model diffusion mass. The value of the Warburg element is $11 \pm 0.3$ Ω.s$^{-0.5}$, which is quite low in comparison with bare electrodes (840 kΩ.s$^{-0.5}$). It translates to the resistivity of the system for any mass transfer [41] (in this case, charge) due to the PEDOT:PSS coating. This result is in excellent agreement with the cut-off frequency variation.

For the segmentation of fast ripples (FRs), we employed the pipeline depicted in Figure 4. This process relies on the visual detection of FRs and subsequent filtering within the frequency band of 200–600 Hz, as explained in detail previously. The analysis of the iEEG recordings shows that the number of FRs varies greatly each day, from 55 on day 2; 336 on day 4; 1791 on day 7; 627 on day 9; and 1713 on day 11. The energy in both FR and background bands was found to be higher for signals recorded with the Au/PEDOT:PSS electrode compared to the Au one, as shown in Figure 7. Both box plots depicted in Figure 7D,E follow the same trend. The energy of the recorded FRs has a similar distribution from day 2 to day 7. However, on day 9, the energy is attenuated on both

electrodes and in both the background and the FRs frequency bands. Considering day 2 as the reference: the median value of the energy for the FR band attenuates by 11% and 10% for Au and for Au with PEDOT:PSS, respectively. In the background band, the attenuation is slightly inferior, with 7% on both electrodes. On day 11, this number decreases even more to 25% and 22% for Au and Au/PEDOT:PSS electrodes, respectively. The equivalent numbers in the background bands are 13% and 12% for Au and Au/PEDOT:PSS, respectively. A one-tailed paired samples *t*-test was performed between the two time series recorded on each day. It revealed statistical significance, with a *p*-value always inferior to $10^{-5}$. Therefore, coating electrodes with PEDOT:PSS makes visual detection easier since it gives signals a higher amplitude. However, the improvement is on all frequencies and not only on the FR band. In the FR band, the difference of energy between the electrodes is 6.3 dB on day 2; 6.1 dB on day 9; and 6.2 dB on day 11, based on the median values. On the background band, the difference in energy is very similar: 6.2 dB on days 2 and 9, and 6.1 dB on day 11. Hence, the energy recorded with the Au/PEDOT:PSS is on average four times higher than with the Au wire, which would match the prediction of the model for the Transfer Function of the system in Figure 6.

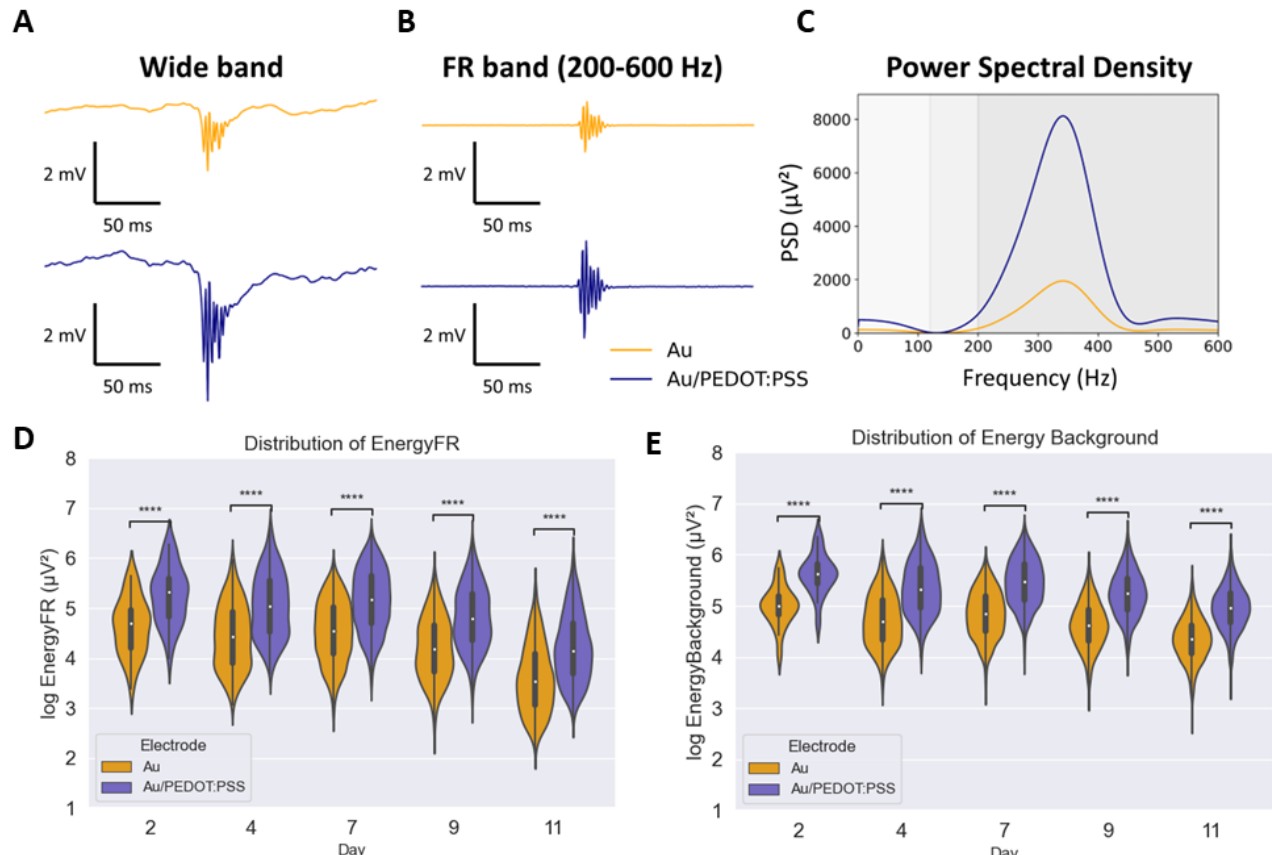

**Figure 7.** Comparison between FRs recorded with uncoated and coated Au/PEDOT:PSS. (**A**): An example of FR recorded from an epileptic mouse in the wide band with Au (top, yellow) and Au/PEDOT:PSS (bottom, blue) electrodes. (**B**): Same signals in the FR band (200–600 Hz). (**C**): Power Spectral Density (PSD) of the segmented FRs. (**D**): Boxplots of the $log_{10}$ of the energy in the FR band of Au and Au/PEDOT:PSS microelectrodes from day 2 to day 11. (**E**): Boxplots of the $log_{10}$ of the energy in the Background band (3.5–200 Hz) of Au and Au/PEDOT:PSS microelectrodes from day 2 to day 11. **\*\*\*\*** indicates *p*-value $10^{-5}$.

## 4. Discussion

PEDOT:PSS-coated electrodes are widely used in modern bioelectronics and neural recordings [27,33,42,43]. They have been referred to as the new golden standard for neu-

roelectronic interfaces [42] since they can improve the SNR ratio during the recording of LFPs [44]. PEDOT:PSS is also characterized by its relatively high biocompatibility (low toxicity) and biostability [45,46]. In the context of epilepsy, they can offer a solution to the FR recording issues related to microelectrodes' high impedance and mechanical mismatch with the brain tissues. However, there is a lack of studies that specifically address the problem of improving FR observability by lowering the impedance of the electrode. In our recent study [17], we investigated, using computational modeling, the impact of PEDOT:PSS voting on the quality of recorded FRs. This study included preliminary in vivo results that showed that PEDOT:PSS can record FRs with higher energy compared to Au and stainless steel wires. Still, the in vivo results were not conclusive since they were based on one recording day, and only 2/3 of the mice presented this improvement in the recorded FR signals. This study aimed to experimentally validate the improvement of FR observability and visual detection using PEDOT: PSS-coated gold wire microelectrodes with an improved experimental scheme.

Regarding FR detection, it should be noted that, although various detectors have been used for the automatic detection of FRs, their outcome is highly dependent on the detector type and parameters. As a result, visual detection of FRs is still the golden standard in the neuroscience field [9]. The results presented in this work indicate that Au/PEDOT:PSS microelectrodes significantly increase the energy of the signal of interest compared to Au microelectrodes throughout the two weeks of postimplantation recording (Figure 7). These findings validate the results presented in the previous study [17] and also suggest that using PEDOT:PSS-coated microelectrodes can improve FRs; observability, and hence, their detection.

Another result that was observed in this work concerned the postimplantation signal energy variation. According to the segmented FRs, the energy of the signals decreased after the seventh recording day (Figure 7D,E). This can be explained by the formation of scar tissue around the electrodes. The insertion of the electrodes in the brain triggers immune responses which encapsulate the probe with a dense scar. Several events are assigned to the foreign body reaction. This includes the insertion trauma, the disruption of the blood–brain barrier, and the presence of the probe itself [47]. The main reason behind this cascade is the mechanical mismatch between the brain tissue (young modulus of 200 to 1500 pa) and the electrodes (for example 50 Gpa for silicon probes). Accordingly, the scar tissue, which encapsulates the electrode, can be as thick as 100 μm [48,49]. It reduces the number of neurons close to the implant and increases the impedance of the electrodes. Several studies, based on in vivo impedance spectroscopy, investigated the properties of this scar tissue [48,50]. In particular, Charkhkar et al. [51] observed that the main increase in the impedance takes place during the first week postimplantation, then it stabilizes for the PEDOT-coated microelectrodes, making it more suitable for chronic implantation compared to the Au one. They attributed this result to improved coupling between microelectrode and brain tissue. This is consistent with our results where we found that the decrease in the signal's energy after day 7 was significantly lower in the case of PEDOT:PSS-coated electrodes. In addition, the ratio of the FRs' energy to the background energy was significantly higher for day 11 ($p < 0.0001$) for the Au/PEDOT:PSS microelectrode compared to the Au one. This suggests that for Au/PEDOT:PSS, the attenuation due to the ETI is lower in the FR band compared to the background. This was expected from the transfer function of the system (Figure 6). However, this transfer function did not take into account the scar tissue layer.

Another point to mention would be that we neglected the wire electrodes' resistivity. This resistivity is due to the insulator surrounding the electrodes in the conductive medium of the brain. Its value is equal to a few pF [34]. Moreover, the impedance of the amplifier was equal to 50 MΩ, according to the technical documentation (Deltamed TM) which we assumed to be infinite. These approximations allowed us to simplify the equivalent circuit since the input of the amplifier is the potential of a capacitance that also accounts for the various parasitic effects in parallel. Nonetheless, the value of the equivalent capacitance

is unknown. In the same plane, the effect of the soldering and the connectors and wires' contribution is not known either. Our observations show a constant difference of 6 dB between the two materials tested in this study. Therefore, one can only speculate that one of the parasitic capacitance is in the order of 1 nF and is predominant compared to the others. Accordingly, our model would portray very well the Transfer Function of the system and could be applied to other materials.

The present study is limited by the number of animals used in the in vivo recordings. In addition, the number of recorded FRs varied greatly between the recording sessions. This can be explained by the inherent variability of the kainate model [37], in which epileptogenesis progresses at a different rate in each animal. Furthermore, the rate of FRs is related to the activity of the animal, which is conditioned by its amount of stress. Nonetheless, the high number of segmented events is sufficient to deduce pertinent results that are in line with our previous findings in [17]. This work proposes new and improved recording microelectrodes specific for the recording of FRs. These PEDOT:PSS-coated electrodes contribute to better FR detection results by increasing their observability.

## 5. Conclusions

One of the main challenges of using microelectrodes is their high impedance that induces distortion and low signal-to-noise ratios. This is particularly problematic in the case of high-frequency events such as FRs. PEDOT:PSS has been widely adopted as one of the best conductive polymers for coating neural interfaces. This study addressed the use of PEDOT:PSS-coated Au electrodes to enhance the quality of recorded FR, and thereupon, their detection. The main conclusions obtained indicates that using Au:PEDOT:PSS microelectrodes results in better FR observability and higher signal energy. This suggests that tuning the impedance of classical Au microelectrodes with the PEDOT:PSS coating can improve FR detection and help to improve the EZ delineation during presurgical evaluation. Future work will focus on the influence of the mechanical properties of the electrode on the intensity of the inflammatory response and scar tissue formation. The electrical properties of gliosis caused by the electrode implantation can be improved via the PEDOT:PSS coating (reduced mechanical mismatch between the microelectrode and the brain tissue) and should be further investigated.

**Author Contributions:** Conceptualization, H.M., G.D. (Gautier Dauly), A.E.M. and M.A.H.; methodology, G.D. (Gautier Dauly), M.A.H., F.W. and H.M.; validation, G.D. (Gabriel Dieuset); investigation, H.M., G.D. (Gautier Dauly) and M.A.H.; data curation, G.D. (Gabriel Dieuset); writing—original draft preparation, H.M., G.D. and M.A.H.; writing—review and editing, H.M., G.D., M.A.H., A.E.M., F.W. and E.I.; visualization, G.D.; supervision, F.W., E.I. and M.A.H.; project administration, F.W.; funding acquisition, F.W. and E.I. All authors have read and agreed to the published version of the manuscript.

**Funding:** This work was supported by the French National Research Agency through the NEURO-SENSE project (ANR-18-CE19-0013).

**Institutional Review Board Statement:** The animal study protocol was approved by the European Union directive in use (Dir 2010/63/UE) and by the ethics committee on animal experimentation of Rennes and received agreement from the French national legal entities (agreement N APAFIS # 2327-2015101914507202).

**Informed Consent Statement:** Not applicable.

**Data Availability Statement:** The data presented in this work will the available upon request.

**Acknowledgments:** E.I. wishes to thank the CMP cleanroom staff at the Center of Microelectronics in Provence for their technical support during the development of the project.

**Conflicts of Interest:** The authors declare no conflicts of interest.

**Abbreviations**

The following abbreviations are used in this manuscript:

| | |
|---|---|
| EZ | Epileptogenic Zone |
| FRs | Fast Ripples |
| Au | Gold |
| PEDOT:PSS | poly(3,4-ethylenedioxythiophene)-poly(styrene sulfonate) |
| Au/PEDOT:PSS | PEDOT:PSS-coated Au |
| LFPs | Local Field Potentials |
| SNR | Signal-to-Noise Ratio |
| CP | Conductive Polymers |
| EDOT | 3,4-ethylenedioxythiophene |
| ETI | Electrode–Tissue Interface |
| LH | Left Hippocampus |
| LR | Right Hippocampus |
| DG | Dentate Gyrus |
| SS | Stainless Steel |
| R | Ripples |
| EIS | Electrochemical Impedance Spectroscopy |
| CV | Cyclic Voltammetry |

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
