# Peer review of "Tuning Microelectrodes’ Impedance to Improve Fast Ripples Recording"

_bioengineering, doi:10.3390/bioengineering11010102_

Round 1
Reviewer 1 Report
Comments and Suggestions for Authors
I think this is an excellent paper and should be published with a few changes, some of which will simply make it more attractive to the reader:
1. Abstract Line #5: "small surface electrodes" Do you mean "physically small electrodes"? Small surface electrodes implies "small surface area", and one of the things that changes is the surface area of the electrode when PEDOT:PSS is applied, so I think “physically small electrodes” may be a better description.
2. Why do you use the terminology “spread resistance” instead of “solution resistance” or “series resistance”? I think “series resistance” might be more appropriate in this context. I have only seen spread resistance used for corrosion studies.
3. Line # 127: Change “reatio” to “ratio”
4. I find it interesting that Cdl increases by 3 orders of magnitude with the use of PEDOT:PSS yet the surface area actually seems to be smaller from the SEM images. Double-layer capacitance, Cdl, is proportional to electrode surface area, so my first impression from the SEM images is that surface roughness has been diminished, not increased. I didn’t attempt to calculate the increase in surface area due to the thickness of the coating, but I doubt it causes 3 orders of magnitude change in the surface area! Some text regarding this issue would be helpful to the reader.
5. Line #196: Reference to “Figure ??.A” needs to be fixed.
6. Line #205: You indicate that the CV curve was taken with the electrodes in an aqueous solution. What solution was used? I assume that some conductive ionic solution was used so it should be described along with the ionic concentration.
7. Figure 6 translates the previous models and results into a Bode plot format. Unfortunately, this plot shows “gain” at DC (0 Hz). That cannot be true, especially for the PEDO:PSS circuit model that you show because there is a series capacitor (Cdl) which will block the flow of any DC current. For the bare gold electrode both Cdl and Q will also block DC. Indeed, Figure 2 shows the asymptotic increase in Z as frequency decreases for both electrode types. I believe that this inconsistency appears because of the capacitive voltage divider that occurs in the models while ignoring the input impedance of the transimpedance amplifier used in the potentiostat. From a quick look at the values, I believe that the low frequency cutoff may be in the 0.1 – 1 Hz range for the PEDOT:PSS electrodes. So, to fix this, I think you can either make sure the models are consistent at low frequencies or truncate the |H|(dB) plot at 10 Hz.
8. Line #231: Change “rised” to “rises”
9. Line #236: Change “hich” to “which”
Author Response
I think this is an excellent paper and should be published with a few changes, some of which will simply make it more attractive to the reader:
We thank the reviewer for the positive feedback and the comments and reviews made.
- Abstract Line #5: "small surface electrodes" Do you mean "physically small electrodes"? Small surface electrodes implies "small surface area", and one of the things that changes is the surface area of the electrode when PEDOT:PSS is applied, so I think “physically small electrodes” may be a better description.
Yes, we mean physically small electrodes, not the effective surface area. "small surface electrodes" were replaced with “physically small electrodes” in the abstract for more precision and clarity. (Line #5).
- Why do you use the terminology “spread resistance” instead of “solution resistance” or “series resistance”? I think “series resistance” might be more appropriate in this context. I have only seen spread resistance used for corrosion studies.
The solution resistivity refers only to the resistance of the electrolyte or solution between the electrode and the tissue which is 75 ohm in the case of biological solution (PBS). In addition, series resistivity is a specific component within an electrical circuit, representing the total resistance encountered in a series of elements. On the other hand, spread resistivity refers to the nature of the surrounding medium (such as electrode geometry and conductivity) and how current spreads through it. Therefore, we used the spreading resistance which reflects both the medium and electrode surface properties.
- Line # 127: Change “reatio” to “ratio”
This was corrected in the manuscript (Line #137).
- I find it interesting that Cdl increases by 3 orders of magnitude with the use of PEDOT:PSS yet the surface area actually seems to be smaller from the SEM images. Double-layer capacitance, Cdl, is proportional to electrode surface area, so my first impression from the SEM images is that surface roughness has been diminished, not increased. I didn’t attempt to calculate the increase in surface area due to the thickness of the coating, but I doubt it causes 3 orders of magnitude change in the surface area! Some text regarding this issue would be helpful to the reader.
Thank you for this interesting comment. The answer to this question lies in the PEDOT:PSS mixed ionic/electronic conductivity with a porous structure that is able to host a high volume of electrical charges. This behavior is known as the volumetric capacitance that increases the double-layer capacitance and therefore is able to improve the electrode’s impedance. It should be mentioned that the effective electrochemical surface area of the electrode changes due to the volumetric capacitive itself. We recently published a work on PEDOT:PSS electropolymerization on various surfaces and addressed this issue. please refer to our recently published work [Mousavi et al. 2023]. We added a few sentences in the manuscript to clarify this point in the methods and results section: (Line #245) “This is due to the increase in the effective electrochemical surface area of the microelectrode after coating.”, (Line #117) “describes the resistance between the working and the counter electrode”. We also added the reference of [Mousavi et al. 2023].
Mousavi, Hajar, et al. "Kinetics and Physicochemical Characteristics of Electrodeposited PEDOT: PSS Thin Film Growth." Advanced Electronic Materials (2023): 2201282.
- Line #196: Reference to “Figure ??.A” needs to be fixed.
This was fixed in the new version of the manuscript.
- Line #205: You indicate that the CV curve was taken with the electrodes in an aqueous solution. What solution was used? I assume that some conductive ionic solution was used so it should be described along with the ionic concentration.
It was taken in PBS solution, which is prepared by the dissolving of phosphate buffer silane in deionized water (Sigma aldrich-P4417). This combination produces 0.01 M phosphate buffer, 0.0027 M potassium chloride, and 0.137 M sodium chloride, pH 7.4, at 25°C. This was added in the manuscript in the methods section 2.3 “cyclic voltammetry measurements” ,“ The volumetric capacitance of the electrodes before and after coating with PEDOT:PSS was measured using the cyclic voltammetry (CV) technique in a three-electrode set-up (Ag/AgCl electrode as the reference and platinum wire as a counter electrode. The CV scans are applied with scan rates of 100 mV/s in a voltage range of -0.9 to 0.6 V in electrolyte solution. The aqueous solution used consisted of a Phosphate Buffered Saline (PBS) of 0.01 M phosphate buffer, 0.0027 M potassium chloride, and 0.137 M sodium chloride (pH 7.4, at 25°C).” (Lines #129-136).
- Figure 6 translates the previous models and results into a Bode plot format. Unfortunately, this plot shows “gain” at DC (0 Hz). That cannot be true, especially for the PEDO:PSS circuit model that you show because there is a series capacitor (Cdl) which will block the flow of any DC current. For the bare gold electrode both Cdl and Q will also block DC. Indeed, Figure 2 shows the asymptotic increase in Z as frequency decreases for both electrode types. I believe that this inconsistency appears because of the capacitive voltage divider that occurs in the models while ignoring the input impedance of the transimpedance amplifier used in the potentiostat. From a quick look at the values, I believe that the low frequency cutoff may be in the 0.1 – 1 Hz range for the PEDOT:PSS electrodes. So, to fix this, I think you can either make sure the models are consistent at low frequencies or truncate the |H|(dB) plot at 10 Hz
Yes, you are right. There is an error in the plot. it actually starts at 1 Hz (10^0) not at 0 Hz. We have corrected this error in the new Figure 6 in the manuscript.
- Line #231: Change “rised” to “rises”
This was corrected in the new version of the manuscript.
- Line #236: Change “hich” to “which”
This was corrected in the new version of the manuscript.

Reviewer 2 Report
Comments and Suggestions for Authors
In this manuscript, the authors proposed a method to tune microelectrodes impedance to improve the ability of fast ripples recording. The proposed electrode surface improved the power in FR band by modifying PEDOT:PSS so that it can be used to the delineation of epileptogenic zones in preoperative assessment.
The manuscript is suggested as major revision, and the specific comments are listed below:
1. The electrode condition and the specific implantation position were not seen in Figure 3, and the resolution of the pictures in the manuscript needs to be improvedï¼›
2. It is recommended to add the electrode manufacturing process diagram as a supplement in 2.1ï¼›
3. In the example data in Figures 4C and 4D, lack explanation and description of data results;
4. The author mentioned that “the band-pass filtering of FRs is misleading since it can lead to misinterpretation of false-Ripples”, and “In the pipeline depicted in Figure 4, this process relies on the visual detection of FRs and subsequent filtering within the frequency band of 200-600 Hz”, so how to avoid the misinterpretation?
5. The author mentioned that “The PEDOT:PSS coating shifted the capacitive behavior of the electrodes to a frequency as low as 63 ± 0.1 Hz” . Does the author have the experimental data to prove the change, and how about the consistency of the phenomenon?
Comments on the Quality of English LanguageThe English is OK.
Author Response
In this manuscript, the authors proposed a method to tune microelectrodes impedance to improve the ability of fast ripples recording. The proposed electrode surface improved the power in FR band by modifying PEDOT:PSS so that it can be used to the delineation of epileptogenic zones in preoperative assessment.
Thank you for the review and the comments provided, we hope we were able to address all the concerns and improve the manuscript.
The manuscript is suggested as major revision, and the specific comments are listed below:
- The electrode condition and the specific implantation position were not seen in Figure 3, and the resolution of the pictures in the manuscript needs to be improvedï¼›
Figure 3 presents the general setup of the experimental protocol. In particular, Figure 3.B depicts a diagram representing the implantation of the electrodes into the hippocampus of the mouse. The exact positions are given in the legend and in the text (methods section): “ In the Right Hippocampus (RH) an Au and an Au/PEDOT:PSS electrodes were inserted at AP=-2.0 mm, ML=-1.5 mm, DV=-1.9 mm. In the Left Hippocampus (LH) two Au/PEDOT:PSS electrodes were inserted at AP=-2.0 mm, ML=+1.5 mm, DV=-1.9 mm. The electrodes placements were determined according to the atlas of the mouse brain (paxinos2019)”
- It is recommended to add the electrode manufacturing process diagram as a supplement in 2.1ï¼›
Since the manufacturing was already described in details in a previous publication we did not include it in this manuscript. However, we added the reference in 2.1 [Mousavi et al. 2023].
Mousavi, Hajar, et al. "Kinetics and Physicochemical Characteristics of Electrodeposited PEDOT: PSS Thin Film Growth." Advanced Electronic Materials (2023): 2201282.
- In the example data in Figures 4C and 4D, lack explanation and description of data results;
Yes, you are right. We added a description of these results in the text (results section). “Figure 4.C depicts the watershed segmentation algorithm results for an event of interest. It portrays the spectrogram of the signal in the FR band. The algorithm allows for the delineation of the FR onset and offset. This delineation is portrayed in Figure4.D in the time domain. The time stamps (onset and offset) of FRs were used to extract the specific energy features to compare between electrode types.” (#Lines 197-201)
- The author mentioned that “the band-pass filtering of FRs is misleading since it can lead to misinterpretation of false-Ripples”, and “In the pipeline depicted in Figure 4, this process relies on the visual detection of FRs and subsequent filtering within the frequency band of 200-600 Hz”, so how to avoid the misinterpretation?
Yes. In this pipeline, we avoided the use of filters since blindly applying filters can result in false ripple selection. More precisely the filtering of sharp transient events such as epileptic spikes or artifacts or signals with harmonics can result in signals resembling ripples as indicated in the work of Benar et al. 2009 [Bénar et al. 2009]. Accordingly, we first visually selected the events of interest (Fast ripples), and then we used wavelets to extract the FR component from the selected signal. Finally, we applied the segmentation method explained in [Al Harrach et al. 2021,2023] to delineate the start (onset) and end (offset) of the FRs. This pipeline allows us to avoid false detection and misinterpretation.
Al Harrach M, Dauly G, Seyedeh-Mousavi H, Dieuset G, Benquet P, Ismailova E, Wendling F. Improving Fast Ripples Recording With Model-Guided Design of Microelectrodes. IEEE Trans Biomed Eng. 2023 Aug;70(8):2496-2505. doi: 10.1109/TBME.2023.3250763.
Harrach MA, Benquet P, Wendling F. Long term evolution of fast ripples during epileptogenesis. J Neural Eng. 2021 Apr 26;18(4). doi: 10.1088/1741-2552/abf774. PMID: 33849005
Bénar CG, Chauvière L, Bartolomei F, Wendling F. Pitfalls of high-pass filtering for detecting epileptic oscillations: a technical note on "false" ripples. Clin Neurophysiol. 2010 Mar;121(3):301-10. doi: 10.1016/j.clinph.2009.10.019. Epub 2009 Dec 1. PMID: 19955019.
- The author mentioned that “The PEDOT:PSS coating shifted the capacitive behavior of the electrodes to a frequency as low as 63 ± 0.1 Hz” . Does the author have the experimental data to prove the change, and how about the consistency of the phenomenon?
Thank you for the interesting comment. Indeed the characterization of electrode performance has been carried out by measuring the impedance electrode. The cutoff frequency refers to the frequency at which the electrode's impedance reaches a certain threshold value, typically -3 decibels (dB) or 70.7% of the DC (direct current) impedance. This phenomenon represents the bode plot representation of impedance with a 45-degree phase showing that electrodes transit from predominantly resistive to predominantly capacitive regime [Mousavi et al. 2023]. Therefore, in our results, the value of 63Hz corresponds to the 45-degree phase of the impedance.
Mousavi, Hajar, et al. "Kinetics and Physicochemical Characteristics of Electrodeposited PEDOT: PSS Thin Film Growth." Advanced Electronic Materials (2023): 2201282.
Reviewer 3 Report
Comments and Suggestions for Authors
In this work, authors employed biophysical modelling to compare Gold (Au) and 8 poly(3,4-ethylenedioxythiophene)-poly(styrene sulfonate) Au coated (Au/PEDOT:PSS) microelectrodes and then implanted in the CA1 hippocampal neural network of epileptic mice to record Fast ripples (FRs) during 10 epileptogenesis. Additionally, efforts were made to demonstrate the effect of microelectrode impedance on signal energy and FR detection. The work is interesting to the readers. However, following points should be addressed before publication:
1) The microelectrode impedance can be tuned by using any conducting materials. Why authors preferred PEDOT:PSS?
2) Electropolymerization was carried out at 1.1 V for 50 s. What if higher voltage is applied for longer duration?
3) PEDOT:PSS is a conducting polymer. The impedance of modified electrode will vary depending on the deposited PEDOT:PSS thin film on Au electrode. What is the thickness of PEDOT:PSS on Au electrode?
4) How does the thickness of conducting material PEDOT:PSS in this manuscript affect the FR observability and detection?
5) As shown in Figure 1, SEM images of the cross-section of gold (Au), and Au coated with PEDOT:PSS (Au:PEDOT/PSS) wire microelectrodes implanted into the right (A) and left hippocampus (B) of the mouse are presented. In Figure A, both Au, Au:PEDOT/PSS are included , however in Figure B both electrodes seems to be Au:PEDOT/PSS. Why the authors did not take same set of electrodes in both LH and RH hippocampus. Please clarify? Also include a Clear SEM image of both Au and Au:PEDOT/PSS.
6) The electrodes were prepared in pairs as portrayed in 1. What does 1 correspond to Figure 1?
7) The SEM micrographs, presented in Figure ??.A, showed that the morphology of PEDOT:PSS coating is highly dependent on the surface roughness. Please include the Figure number in line number 196. Morphology depend on Au electrode surface? Sentences in line 195-198 seems confusing. Please check carefully and modify accordingly.
8) For recording CV, authors used two electrode system working electrode (WE) and reference electrode (RE) or 3 electrode system WE, counter electrode (CE), and RE
9) Please include figure number in line 210.
10) Please comment on and discuss the cytotoxicity of PEDOT:PSS modified Au electrode in the manuscript as it is critical for in-vivo applications.
11) The concluding remarks are missing in the manuscript. Authors should include the conclusions in a separate section.
Author Response
In this work, authors employed biophysical modelling to compare Gold (Au) and 8 poly(3,4- ethylenedioxythiophene)-poly(styrene sulfonate) Au coated (Au/PEDOT:PSS) microelectrodes and then implanted in the CA1 hippocampal neural network of epileptic mice to record Fast ripples (FRs) during 10 epileptogenesis. Additionally, efforts were made to demonstrate the effect of microelectrode impedance on signal energy and FR detection. The work is interesting to the readers. However, following points should be addressed before publication:
We thank the reviewer for the comments and the feedback.
1) The microelectrode impedance can be tuned by using any conducting materials. Why authors preferred PEDOT:PSS?
Regarding the impedance reduction, several organic coatings have been suggested such as carbon nanotube, Polypyrol and PEDOT (with various counter ions such as PSS) polymer. Among all available coating, PEDOT:PSS seems to be the best option so far thanks to its biocompatibility, high volumetric capacitance as well as facile coating.
2) Electropolymerization was carried out at 1.1 V for 50 s. What if higher voltage is applied for longer duration?
The electropolymerization of PEDOT:PSS can be carried out in the voltage window between 0.9 V to 1.2 V in an aqueous media This voltage range avoids water hydrolysis during electropolymerization., The important factor is to achieve a stable coating without any cracks and delamination because of its weak adhesion forces and the absence of any chemical linkage with the substrate. We previously showed that the increase in thickness reduces the film stability according to the surface properties [Mousavi et al. 2023]. In our work, we observed that before 50 seconds, the surface is not fully coated with PEDOT:PSS, and after 50s, we can see the cracks and delamination of PEDOT:PSS appears.
Mousavi, Hajar, et al. "Kinetics and Physicochemical Characteristics of Electrodeposited PEDOT: PSS Thin Film Growth." Advanced Electronic Materials (2023): 2201282.
3) PEDOT:PSS is a conducting polymer. The impedance of modified electrode will vary depending on the deposited PEDOT:PSS thin film on Au electrode. What is the thickness of PEDOT:PSS on Au electrode?
This is indeed useful to know but due to the electrode form, it’s not possible to measure as electrodes are in the form of microelectrodes. For example, in the case of planar electrodes, a profilometer or SEM can be used to measure the thickness. Please refer to our recently published paper where the kinetics of the thickness growth was investigated on macroscale Au electrodes [Mousavi et al.2023]. We believe that the thickness should be around 100-500 nm.
Mousavi, Hajar, et al. "Kinetics and Physicochemical Characteristics of Electrodeposited PEDOT: PSS Thin Film Growth." Advanced Electronic Materials (2023): 2201282.
4) How does the thickness of conducting material PEDOT:PSS in this manuscript affect the FR observability and detection?
Due to the technical complexity, we did not measure the thickness. However, the reduction of impedance caused by PEDOT:PSS coating shifts the capacitive behavior in the range of FRs together with a reduction of the impedance. It results in a higher signal-to-noise ratio and avoids signal distortion due to the phase shift of impedance caused by the capacitive behavior of electrodes.
5) As shown in Figure 1, SEM images of the cross-section of gold (Au), and Au coated with PEDOT:PSS (Au:PEDOT/PSS) wire microelectrodes implanted into the right (A) and left hippocampus (B) of the mouse are presented. In Figure A, both Au, Au:PEDOT/PSS are included , however in Figure B both electrodes seems to be Au:PEDOT/PSS. Why the authors did not take same set of electrodes in both LH and RH hippocampus. Please clarify? Also include a Clear SEM image of both Au and Au:PEDOT/PSS.
-The choice has been made to have enough statistics on PEDOT: PSS-coated electrodes as we have previously studied the FR recording with gold electrodes. However, for this particular work, we only processed the signals obtained from the right hippocampus that was epileptic (after Kainate acid injection). The other two Au:PEDOT/PSS electrodes were only used as control and the corresponding signals were not processed.
-We replaced figure 5.A with clear SEM images of both Au and AU:PEDOT/PSS
6) The electrodes were prepared in pairs as portrayed in 1. What does 1 correspond to Figure 1?
Yes. sorry for the confusion we corrected this in the manuscript.
7) The SEM micrographs, presented in Figure ??.A, showed that the morphology of PEDOT:PSS coating is highly dependent on the surface roughness. Please include the Figure number in line number 196. Morphology depend on Au electrode surface? Sentences in line 195-198 seems confusing. Please check carefully and modify accordingly.
-“Figure ??.A” was replaced by Figure 5.A in the manuscript.
- We checked and improved the sentence in the manuscript. “Figure 5.A show the morphology of the electrode's surface before and after PEDOT:PSS coating. The resulting PEDOT:PSS roughness is due to the surface structure of the Au electrode and its initial roughness”
8) For recording CV, authors used two electrode system working electrode (WE) and reference electrode (RE) or 3 electrode system WE, counter electrode (CE), and RE
It has been measured in three-electrode set-ups. This was added to the manuscript in the methods and result sections. We also added a subsection in the materials and methods (Lines #129-136) to describe the CV measurements. “The volumetric capacitance of the electrodes before and after coating with PEDOT:PSS was measured using the cyclic voltammetry (CV) technique in a three-electrode set-up (Ag/AgCl electrode as the reference and platinum wire as a counter electrode. The CV scans are applied with scan rates of 100 mV/s in a voltage range of -0.9 to 0.6 V in electrolyte solution. The aqueous solution used consisted of a Phosphate Buffered Saline (PBS) of 0.01 M phosphate buffer, 0.0027 M potassium chloride, and 0.137 M sodium chloride (pH 7.4, at 25°C)”
9) Please include figure number in line 210.
The figure number was added in the manuscript.
10) Please comment on and discuss the cytotoxicity of PEDOT:PSS modified Au electrode in the manuscript as it is critical for in-vivo applications.
PEDOT:PSS is an excellent choice as it has mixed ionic and electronic conductivity to facilitate communication with the brain. In addition, it has shown relatively high biocompatibility, making it a popular choice for various biomedical applications [Williamson et al. 2015, Galliani et al. 2023]. The biocits ompatibility of PEDOT:PSS refers to its low toxicity in direct contact with living cells and tissues without causing significant adverse effects, biostability in physiological environments, maintaining its structural and functional integrity over time, and tissue biocompatibility (absence of strong immune responses or adverse reactions when in contact with living tissues). We added this and the references in Lines #290-291.
Williamson A, Ferro M, Leleux P, Ismailova E, Kaszas A, Doublet T, Quilichini P, Rivnay J, Rózsa B, Katona G, Bernard C, Malliaras GG. Localized Neuron Stimulation with Organic Electrochemical Transistors on Delaminating Depth Probes. Adv Mater. 2015 Aug;27(30):4405-4410. doi: 10.1002/adma.201500218.
Galliani M, Ferrari LM, Bouet G, Eglin D, Ismailova E. Tailoring inkjet-printed PEDOT:PSS composition toward green, wearable device fabrication. APL Bioeng. 2023 Jan 3;7(1):016101. doi: 10.1063/5.0117278.
11) The concluding remarks are missing in the manuscript. Authors should include the conclusions in a separate section.
Thank you for this remark. We added a conclusion section in the new version of the manuscript (Lines #357-371).
“One of the main challenges of using microelectrodes is the high impedance that induces distortion and low signal-to-noise ratios. This is particularly problematic in the case of high-frequency events such as FRs. PEDOT:PSS has recently been widely adopted as one of the best conductive polymers for coating neural interfaces. This study addressed the use of PEDOT/PSS coated Au electrodes to enhance the quality of recorded FR and thereupon their detection. The main conclusions obtained indicates that using Au:PEDOT/PSS microelectrodes results in better FR observability and higher signal energy. This suggests that Tuning the impedance of classical Au microelectrodes with PEDOT/PSS coating can improve FR detection and help improve EZ delineation during presurgical evaluation. Future work will focus on the influence of the mechanical properties of the electrode on the intensity of the inflammatory response and scar tissue formation. The electrical properties of gliosis caused by the electrode implantation can be improved through PEDOT:PSS coating (reduced mechanical mismatch between the microelectrode and the brain tissue) and should be further investigated.
Round 2
Reviewer 2 Report
Comments and Suggestions for Authors
References from previous work have been added, and the missing data analysis explanations have been added to the results.
I suggest accpetance.
Comments on the Quality of English LanguageEnglish is OK.
Reviewer 3 Report
Comments and Suggestions for Authors
The authors have made significant changes to the manuscript.